# Biological Efficacy of Cochlioquinone-9, a Natural Plant Defense Compound for White-Backed Planthopper Control in Rice

**DOI:** 10.3390/biology10121273

**Published:** 2021-12-04

**Authors:** Yoon-Hee Jang, Sopheap Yun, Jae-Ryoung Park, Eun-Gyeong Kim, Byoung-Ju Yun, Kyung-Min Kim

**Affiliations:** 1Division of Plant Biosciences, School of Applied Biosciences, College of Agriculture and Life Sciences, Kyungpook National University, Daegu 41566, Korea; uniunnie@naver.com (Y.-H.J.); icd92@naver.com (J.-R.P.); dkqkxk632@naver.com (E.-G.K.); 2Graduate School of Science, Royal University of Phnom Penh, Sangkat Teuk Laak 1, Russian Federation Boulevard, Toul Kork, Phnom Penh 12101, Cambodia; yun.sopheap@rupp.edu.kh; 3School of Electronics Engineering, College of IT Engineering, Kyungpook National University, 80, Daehak-ro, Buk-gu, Daegu 41566, Korea

**Keywords:** rice, biotic stress, white-backed planthopper, secondary metabolite, cochlioquinone-9

## Abstract

**Simple Summary:**

This study investigated the biological efficacy of cochlioquinone-9 (cq-9), a plant secondary metabolite, for controlling white-backed planthopper (WBPH) and compared the gene expression levels following cq-9 treatment. The results show that cq-9 enhances plant growth against WBPH and is associated with aromatic amino acid-related plant defense genes. This demonstrates the potential of cq-9 to replace chemical pesticides and suggests a new method for controlling WBPH.

**Abstract:**

Rice is exposed to various biotic stresses in the natural environment. The white-backed planthopper (*Sogatella furcifera*, WBPH) is a pest that causes loss of rice yield and threatens the global food supply. In most cases, pesticides are used to control WBPH. However, excessive use of pesticides increases pesticide resistance to pests and causes environmental pollution. Therefore, it is necessary to develop natural product-based pesticides to control WBPH. Plants produce a variety of secondary metabolites for protection. Secondary metabolites act as a defense against pathogens and pests and are valuable as pesticides and breeding materials. Cochlioquinone is a secondary metabolite that exhibits various biological activities, has a negative effect on the growth and development of insects, and contributes to plant defense. Here, we compared plant growth after treatment with cochlioquinone-9 (cq-9), a quinone family member. cq-9 improved the ability of plants to resist WBPH and had an effect on plant growth. Gene expression analysis revealed that cq-9 interacts with various defense-related genes to confer resistance to WBPH, suggesting that it is related to flavonoid compounds. Overall, this study provides insight into the mechanisms of WBPH resistance and suggests that cq-9 represents an environmentally friendly agent for WBPH control.

## 1. Introduction

Rice is an important staple food that is grown worldwide in a variety of climates [1]. However, the yield of rice is affected by several biotic and abiotic factors. A significant yield of rice is lost each year because of biological stress, of which approximately 25% is caused by pests [2]. White-backed planthopper (*Sogatella furcifera*, WBPH) penetrates plant cells and ingests a large amount of nutrients and moisture. When a large number of WBPHs suck in their juice, the leaves dry out and the plant wilts, causing a hopper burn effect [3]. In addition, it reduces the vitality of the rice, delays tillering, and causes stunting, which ultimately leads to the death of rice [4].

In general, pesticides are used to control WBPH, but this is not ideal for several reasons, including the environmental impact and the emergence of pesticide-resistant insects [5]. Alternatively, the use of eco-friendly biological pesticides or the development and cultivation of resistant cultivars are economical and environmentally sound approaches to reduce the damage to the white-backed beetle [6]. This may be achieved using secondary metabolites, which are produced by plants and serve as a defense against pests and pathogens [7]. The secondary metabolites are under genetic control, and in most cases, their production is the result of several enzymatic steps along various biosynthetic pathways [8].

Another defense mechanism of plants is to develop resistance to environmental stress by harboring endophytes [9]. Endophytes are fungi that do not cause disease in plants and are symbiotic [10]. In particular, endophytes that inhabit plant leaves limit the penetration of pathogenic microorganisms into leaf tissues also secrete secondary metabolites that are toxic to plant predators [11]. Fungal endophytes secrete secondary metabolites from host plants, which exhibit antifungal and antibacterial properties even at low concentrations [12]. Therefore, substances secreted by endophytes are associated with host plant ecology and defense mechanisms against pathogens and pests [9]. Cochlioquinone is a major fungal phytotoxin associated with serious crop disease [9]. It is a yellow pigment originally isolated from *cochliobolus miyabeanus*, a parasitic fungus of rice. Cochlioquinone and its derivatives exhibit various biological activities resulting from their side-chain structures [13].

Pest resistance is generally achieved through one or a combination of the following three defenses: antixenosis, which prevents migration or oviposition by driving away or repelling insects; antibiosis, which reduces insect survival, growth, or reproduction after ingestion of tissues; and increased tolerance [14]. Identification of genetic resistance mechanisms will improve our ability to manage WBPH [15]. As sophisticated analytical and molecular genetic tools for functional studies of resistance become more widely available, an approach based on understanding the mechanisms of plant resistance will be an efficient strategy for the development of resistant cultivars for crop protection [16]. The purpose of this study is to evaluate the potential of cq-9 as an eco-friendly agricultural material by measuring the biological efficacy. We measured the WBPH resistance of cq-9 and identified the defense mechanism of cq-9 against WBPH by examining the expression patterns of major genes related to plant defense associated with cq-9 treatment. In addition, we evaluated cq-9 as a useful physiologically active substance. It may be used as a plant-derived eco-friendly substance, and it contributes to the increased yield of crops by suppressing WBPH. It will also be useful for the discovery of new genetic resources.

## 2. Materials and Methods

### 2.1. Plant Material and Field Design

The Cheongcheong/Nagdong Doubled Haploid (CNDH) population was developed by anther culture of F_1_ derived from a cross between Cheongcheong and Nagdong. Cheongcheong is an indica type that is resistant to WBPH, and Nagdong is a japonica type that is susceptible to WBPH. The CNDH population was cultivated for more than 10 years and the agricultural traits are completely fixed. Cheongcheong, Nagdong, CNDH3, and CNDH42-2, which are resistant WBPH, and CNDH45 and Taichung Native 1 (TN1), which are susceptible to WBPH, were used as plant materials. Seeds were obtained by the following method. Rice seeds were soaked in April 2016 and sterilized in water treated with 500 μL per 1 L of spotack pesticide (HANKOOKSAMGONG, Seoul, Korea) at 25 °C for 4 days in the dark and then sown. The seedlings were transplanted to an experimental field of Kyungpook National University in Gunwi in May 2016 and the planting distance was 30 × 15 cm. Fertilizer was applied at a rate of N–P_2_O_5_–K_2_O = 9–4.5–5.7 kg/10a, and the seeds were harvested in October 2016.

### 2.2. White-Backed Planthopper Rearing

The WBPH were reared in a cage maintained at a temperature of 27 ± 1 °C, a humidity of 60–70 %, and a light intensity of 2000 lux for 16 h/day. The breeding cage was constructed with an acrylic plate (50 × 50 × 40 cm), a 100 μm mesh net was used on the back for ventilation, and a sliding door was installed on the front to facilitate feeding. Chucheong, the most preferred food for the white-backed planthopper, was supplied as food and replenished once a week. The WBPH were separated and reared in breeding cages according to growth stage. Usually, they become 1st instars after 9–10 days from eggs, and 2nd and 3rd instars after 14 days. WBPH from this period was used for the experiments.

### 2.3. Extraction of cq-9

The leaves of rice infected with WBPH were collected and ground using liquid nitrogen. Methanol (70%) was added at 10 times the sample amount and incubated at 20 °C and 130 rpm overnight in a shaking incubator. Impurities were filtered out using filter paper (Whatman Grade 2 Qualitative Filter Paper Diameter: 15.0 cm; pore size: 8 μL). To remove the lipid component and non-polar impurities of the methanol extract, n-hexane was used. An equal volume of n-hexane was combined with the filtered extract and added to a separatory funnel and shaken, and the lower layer was collected after separation. The extract was concentrated using a rotary evaporator. Concentration was done in a water bath at 30 °C with cooling water at −3 °C and a speed of 6–8. The presence of the cq-9 band was confirmed using TLC silica gel 60F_254_ plate (Merck, KGaA, Darmstadt, Germany) and a mobile phase of chloroform:methanol:1-butanol:water at a 4:5:6:4 ratio. The Rf value of the separated compound was measured by dividing the distance traveled by the compound relative to the distance of the mobile phase. The band was scraped off from the TLC plate, and cq-9 was isolated and dissolved in 20 mL of 4% methanol in 100% acetone.

### 2.4. Evaluation of the Effect of cq-9 on Rice

Seeds of Cheongcheong, Nagdong, and TN1 were soaked at 25 °C for 4 days. Germinated seeds were sown in plastic boxes (20 × 14 × 4.5 cm) at a planting distance of 2.0 × 0.5 cm. The control group was not treated with any substances, and the experimental group was sprayed with cq-9. Then, 2–3 instars of WBPH were inoculated, and the degree of resistance was scored for each cultivar on days 7, 14, and 21 after sowing. The scoring method measured the degree of damage to the plant using the Standard Evolution System (SES) [17]. The chlorophyll concentration of the rice leaves was measured using a portable chlorophyll meter (SPAD-502, Minolta Camera Co. Ltd., Osaka, Japan). Seedlings of Cheongcheong, Nagdong, TN1, CNDH3, CNDH42-2, and CNDH45 were used for growth comparison following cq-9 treatment. The control group, WBPH inoculation group, cq-9 treatment group, and WBPH inoculation group were treated with cq-9 and grown in each cage. Plant heights were measured 1, 2, and 3 days after treatment. The height of the plant was measured as the shortest length from the point of the stem where the root grows to the highest point of the leaf.

### 2.5. DNA Extraction

The young leaves at the seedling stage were placed into 2 mL tubes, rapidly frozen in liquid nitrogen, and ground using a TissueLyser (QIAGEN, Cat. No. 85220, Hilden, Germany). Then, 700 µL of DNA extraction buffer (2% CTAB, 0.1 M Tris, pH 8.0, 1.4 M NaCl, 1% polyvinylpyrrolidone) was added to the ground sample. After vortexing, the sample was incubated for 20 min in a water bath at 65 °C. After the reaction, 750 µL of PCI (phenol:chloroform:isoamyl alcohol; 25:24:1) was added. The mixture was inverted at room temperature for 20 min and centrifuged at 14,000 rpm at 4 °C for 10 min. After centrifugation, 500 µL of the supernatant was transferred to a 1.5 mL tube, and 350 µL of isopropanol was added. The tube was inverted for 5 min and incubated at −72 °C for 2 min. After dissolving at room temperature, the samples were centrifuged at 14,000 rpm for 10 min. The supernatant was carefully removed and the pellet was washed with 70% ethanol. The washed pellet was centrifuged at 13,000 rpm for 1 min, the solution was removed, and the washing process was repeated. After drying at room temperature, the DNA was dissolved in 50 µL of ddH_2_O. The DNA concentration was measured using a NanoDrop 2000 Spectrophotometer (ND-2000; Nanodrop, Waltham, MA, USA).

### 2.6. RNA Extraction

Total RNA was extracted from leaf tissue at the seedling stage of rice using the RNase Mini Kit (Qiagen, Cat. No. 74904, Hilden, Germany) according to the manufacturer’s instructions. The young leaves were ground in liquid nitrogen using a mortar and pestle. The ground sample was suspended in 450 µL of RLT buffer containing β-mercaptoethanol and vortexed. The lysate was transferred to a QIAshred spin column, placed in a 2 mL collection tube, and centrifuged at 13,000 rpm for 1 min. The supernatant of the flow-through was transferred to a new 1.5 mL tube and 0.5 volume of ethanol (99%) was added and mixed immediately by pipetting. Then, 650 µL of mixture was transferred to a QIAshredder spin column and centrifuged for 1 min. The column was washed with 700 µL of RW1 buffer and centrifuged for 1 min. The flow-through was discarded and the column was washed with 500 µL of RPE buffer and centrifuged at 13,000 rpm for 1 min. The flow-through was discarded, and 500 µL of RPE buffer was added to an RNase spin column and centrifuged for 1 min to remove the residual liquid from the column. The RNA was eluted in RNase free water (30 µL) and centrifuged for 1 min. The RNA concentration was measured using a NanoDrop 2000 Spectrophotometer.

### 2.7. Quantitative Real-Time PCR (qPCR) Analysis

The qRCRBIO cDNA Synthesis kit (Cat No. PB30.11-10, PCRBIOSYSTEM, Wayne, PA, USA) was used for cDNA synthesis. Quantitative real-time PCR was performed using the Eco Real-Time PCR System. The following were mixed in a reaction tube: 4.0 µL of 5x cDNA synthesis Mix buffer, 1.0 µL of RTase, 100 ng of RNA, and PCR-grade dH_2_O up to a total volume of 20 µL. The mixture was thoroughly mixed by pipetting and incubated at 42 °C for 30 min in a water bath. The cDNA synthesis reaction was used as a template for qPCR analysis. *OsActin* was used as a reference gene and the primers for *OsActin* were forward 5′-ATCCTTGTATGCTAGCGGTCGA-3′ and reverse 5′-ATCCAACCGGAGGATAGCATG-3′. The qPCR conditions were as follows: pre-denaturation at 95 °C for 2 min, followed by 40 cycles of denaturation at 95 °C for 10 s, annealing at 60 °C for 30 s, extension at 72 °C for 15 s, and a final melting curve at 95 °C for 15 s, 55 °C for 15 s, and 95 °C for 15 s.

### 2.8. Statistical SPSS Analysis

All experiments of each section were repeated three times, and the data collected from each replicate were pooled together. The statistical analysis was performed using the SPSS program (IMMSPSS Statistics, version 22, IBMSPSS Statistics, version 22, Redmond, WC, USA). The data were analyzed using two-way ANOVA followed by Duncan multiple range test (significant difference: *p* ˂ 0.05). The data were graphically presented, and the table is presented as mean and standard deviation values.

## 3. Results

### 3.1. Identification of cq-9 and Comparison of Extraction Amount

Thin-layer chromatography (TLC) was performed to identify and separate cq-9 from the methanol extract obtained from TN1, Cheongcheong, and Nagdong (Figure 1A). A band was identified at the same position compared with cq-9 as a control. The retention factor (Rf) value of cq-9 was 0.43 ± 0.01, and the Rf value of Cheongcheong, Nagdong, and TN1 was 0.42 ± 0.01, indicating no difference (Table 1). The concentration of cq-9 separated from silica gel was 7.81 ± 0.29 ng/g in Cheongcheong, 5.43 ± 0.21 ng/g in Nagdong, and 9.67 ± 1.17 ng/g in TN1, which was the highest value in the order TN1, Cheongcheong, and Nagdong from the fresh weight of rice (Figure 1B).

### 3.2. WBPH Resistance of cq-9

To investigate WBPH resistance to cq-9, bio-scoring values were evaluated at 1, 2, and 3 weeks after WBPH inoculation of Cheongcheong, Nagdong, and TN1 (Figure 2A). During the 1 week after inoculation, the bio-scoring value of Cheongcheong was 1.2 ± 0.6, Nagdong 1.3 ± 0.7, and TN1 1.4 ± 1.0. When cq-9 was treated, the bio-scoring value of Cheongcheong was 1.2 ± 0.6, Nagdong 2.6 ± 1.7, and TN1 1.5 ± 1.3. None of the three cultivars were damaged by WBPH. Two weeks after inoculation, the bio-scoring value of Cheongcheong was 3.2 ± 0.6, Nagdong 5.2 ± 0.6, and TN1 3.7 ± 1.0. When cq-9 was treated, the bio-scoring value of Cheongcheong was 1.4 ± 0.0, Nagdong was 3.0 ± 1.2, and TN1 was 2.2 ± 1.0. After 2 weeks of inoculation, when Cheongcheong was treated with cq-9, the bio-scoring value was lower compared with that of the control group. This indicates that when cq-9 was treated, Cheongcheong received less damage from WBPH. Three weeks after inoculation, the bio-scoring value of Cheongcheong was 5.0 ± 0.0, Nagdong 6.1 ± 1.2, and TN1 7.9 ± 1.4. When cq-9 was treated, the bio-scoring value of Cheongcheong was 1.6 ± 0.8, Nagdong 4.7 ± 1.6, and TN1 4.4 ± 1.5. After 3 weeks of inoculation, when Cheongcheong and TN1 were treated with cq-9, the bio-scoring value was lower compared with that of the control group, but there was no difference for Nagdong. In addition, the chlorophyll content, determined by SPAD measurement, was measured at 3 weeks after inoculation with WBPH (Figure 2B). Chlorophyll content of Cheongcheong was 25.3 ± 2.5 SPAD and 26.5 ± 4.8 SPAD following cq-9 treatment, which showed no difference. Chlorophyll content of Nagdong was 14.4 ± 6.2 SPAD and 17.7 ± 11.1 SPAD following cq-9 treatment, which also showed no difference. The chlorophyll content of TN1 was 8.0 ± 0.8 SPAD and 26.4 ± 7.4 SPAD with cq-9 treatment; thus there was a difference in chlorophyll content following cq-9 treatment (Table 2).

### 3.3. Comparison of the Effect of cq-9 on Plant Growth

Seedlings of Cheongcheong, CNDH3, CNDH43, Nagdong, TN1, and CNDH45 were treated with cq-9, and plant lengths were measured after 1, 2, and 3 days. Additionally, plant lengths were measured 1, 2, and 3 days after seedlings were inoculated with WBPH and without cq-9 treatment and after seedlings were inoculated with WBPH after cq-9 treatment (Figure 3). The length of Cheongcheong was 13.4 ± 0.4 cm on the 1st day, 14.5 ± 0.2 cm on the 2nd day, and 16.3 ± 0.5 cm on the 3rd day. The length of Cheongcheong treated with cq-9 was 13.8 ± 0.3 cm on the 1st day, 15.3 ± 0.5 cm on the 2nd day, and 16.9 ± 0.4 cm on the 3rd day. Cheongcheong treated with cq-9 was taller compared with the control group. The length of the Cheongcheong inoculated with WBPH was 13.4 ± 0.3 cm on the 1st day, 14.9 ± 0.0 cm on the 2nd day, and 16.0 ± 0.3 cm on the 3rd day. The length of Cheongcheong inoculated with WBPH after cq-9 treatment was 13.5 ± 0.3 cm on the 1st day, 16.4 ± 0.2 cm on the 2nd day, and 16.7 ± 0.4 cm on the 3rd day. There was no difference on the 1st day after the inoculation with WBPH and the cq-9-treated Cheongcheong was taller on the 2nd and 3rd days. The length of CNDH3 was 14.5 ± 0.2 cm on the 1st day, 15.8 ± 0.7 cm on the 2nd day, and 15.9 ± 0.3 cm on the 3rd day. The length of CNDH3 treated with cq-9 was 11.7 ± 0.5 cm on the 1st day, 16.1 ± 0.5 cm on the 2nd day, and 15.8 ± 0.7 cm on the 3rd day. CNDH3 treated with cq-9 on the 1st day was shorter and there was no difference on the 2nd or 3rd days. The length of CNDH3 inoculated with WBPH was 13.7 ± 0.4 cm on the 1st day, 15.8 ± 0.5 cm on the 2nd day, and 15.7 ± 0.4 cm on the 3rd day. The length of CNDH3 inoculated with WBPH after cq-9 treatment was 11.8 ± 0.1 cm on the 1st day, 16.4 ± 0.5 cm on the 2nd day, and 15.9 ± 0.6 cm on the 3rd day. CNDH3 treated with cq-9 was shorter on the 1st day after inoculation with WBPH and taller on the 2nd day after inoculation. There was no difference on the 3rd day. The length of CNDH42-2 was 16.1 ± 0.2 cm on the 1st day, 14.5 ± 0.6 cm on the 2nd day, and 16.6 ± 0.1 cm on the 3rd day. The length of CNDH42-2 treated with cq-9 was 14.0 ± 0.7 cm on the 1st day, 15.1 ± 0.5 cm on the 2nd day, and 16.6 ± 0.1 cm on the 3rd day. CNDH42-2 treated with cq-9 was shorter on the 1st day but larger on the 2nd day. There was no difference on the 3rd day. The length of CNDH42-2 inoculated with WBPH was 10.9 ± 0.5 cm on the 1st day, 15.4 ± 0.5 cm on the 2nd day, and 16.3 ± 0.2 cm on the 3rd day. The length of CNDH42-2 inoculated with WBPH after cq-9 treatment was 15.6 ± 0.5 cm on the 1st day, 15.6 ± 0.5 cm on the 2nd day, and 15.7 ± 0.5 cm on the 3rd day. CNDH42-2 treated with cq-9 was longer on the 1st day after inoculation with WBPH, and there was no difference on the 2nd and 3rd days. The length of Nagdong was 15.7 ± 1.5 cm on the 1st day, 15.8 ± 1.5 cm on the 2nd day, and 15.9 ± 0.6 cm on the 3rd day. The length of Nagdong treated with cq-9 was 15.4 ± 0.5 cm on the 1st day, 16.7 ± 0.5 cm on the 2nd day, and 15.5 ± 0.5 cm on the 3rd day. There was no difference with respect to cq-9 treatment. The length of the Nagdong inoculated with WBPH was 15.3 ± 1.1 cm on the 1st day, 15.8 ± 0.5 cm on the 2nd day, and 15.4 ± 1.0 cm on the 3rd day. The length of Nagdong inoculated with WBPH after cq-9 treatment was 15.8 ± 0.7 cm on the 1st day, 16.1 ± 0.6 cm on the 2nd day, and 16.0 ± 0.6 cm on the 3rd day. Nagdong treated with cq-9 was longer on the 2nd day after inoculation with WBPH, and there was no difference between the 1st and 3rd days. The length of TN1 was 17.1 ± 0.8 cm on the 1st day, 17.4 ± 0.4 cm on the 2nd day, and 17.5 ± 0.4 cm on the 3rd day. The length of TN1 treated with cq-9 was 16.6 ± 0.5 cm on the 1st day, 17.2 ± 0.4 cm on the 2nd day, and 17.0 ± 0.4 cm on the 3rd day. There was no difference with respect to cq-9 treatment. The length of TN1 inoculated with WBPH was 15.8 ± 0.6 cm on the 1st day, 15.9 ± 0.6 cm on the 2nd day, and 15.9 ± 0.6 cm on the 3rd day. The length of TN1 inoculated with WBPH after cq-9 treatment was 15.6 ± 0.6 cm on the 1st day, 15.8 ± 0.6 cm on the 2nd day, and 17.8 ± 0.1 cm on the 3rd day. There was no difference on the 1st and 2nd day after inoculation with WBPH and cq-9-treated TN1 was taller on the 3rd day (Table 3).

### 3.4. Comparison of Relative Gene Expression Levels of Plant Defense Genes

The relative expression level of plant defense-related genes oryza sativa flavanone 3-hydroxylase (OsF3H1), oryza sativa chorismate mutas (OsCM), OsWRKY45, and oryza sativa non-expressor of pathogenesis-related genes1 (OsNPR1) were compared following treatment with cq-9 and WBPH in Cheongcheong, CNDH3, and CNDH42-2, which are resistant to WBPH, and Nagdong, TN1 and CNDH45, which are susceptible to WBPH (Figure 4). OsF3H1 exhibited similar expression on 1 day after inoculation, whereas the expression in CNDH3 was higher with cq-9 treatment compared with the control group. After 2 days of inoculation, the expression level was significantly higher in plants treated with cq-9 and cq-9 + WBPH in TN1. The expression level was decreased on the 3rd day, but it was significantly higher. OsCM exhibited similar expression levels and 2 days after inoculation, the relative expression levels of OsCM were higher in the plants treated with cq-9 + WBPH in CNDH3 and CNDH45. After 3 days of inoculation, the expression levels were higher following inoculation with WBPH in Cheongcheong. For CNDH3, expression levels were higher in both cq-9-treated and inoculated with WBPH. OsCM exhibited higher expression in CNDH45 when treated with cq-9. OsWRKY showed a similar expression 1 day after inoculation and the relative expression of cq-9-treated plants in Nagdong and TN1 was significantly higher 2 days after inoculation. Three days after inoculation, the relative expression of OsWRKY was higher following both cq-9 treatment and WBPH inoculation in CNDH42 and CNDH45. The expression of OsNPR1 was increased in CNDH42-2 and CNDH45 3 days after inoculation, and the relative expression level was significantly higher following treatment with cq-9.

## 4. Discussion

Rice is a nutritious food for many phytophagous insects, and although hundreds of insects damage rice in a variety of ways, only approximately 20 major species regularly cause significant damage [18]. WBPHs are widely distributed in South Asia, Southeast Asia, and Australia, and they winter in tropical and subtropical regions. Adult WBPHs transport the virus from south to north through long-distance migration in spring, transmit the virus to rice seedlings in newly settled areas, and lay eggs [4,19]. WBPH causes significant damage by transmitting SRBSDV to crops, such as rice, wheat, and maize [20]. Both the nymphs and adults of WBPH eat the stem and leaves of the plant and suck the sap from the plant [21]. As a counterattack, plants have various defense mechanisms to protect themselves [22]. Plants use phytochemicals to reduce the preference for herbivores and reduce pathogen infection. On the other hand, in the induced resistance system, invasion by herbivores and infection by pathogens cause physiological changes in plants, thereby improving plant defense capabilities [23].

Over the past few decades, fewer than 30 cochlioquinones have been isolated from phytopathogenic fungi, primarily from the genera *Bipolaris* and *Stachybotrys* [24]. Cochlioquinone A isolated from *Drechslera sacchari* acts as a specific inhibitor of diacylglycerol kinase, which regulates the activity of protein kinase C [25]. Cochlioquinone A isolated from the leaves of *Vernonia polyanthes* has been shown to inhibit parasites [26]. Cochlioquinone A isolated from the *Helminthosporium sativum* exhibited nematode activity [27]. cq-9 isolated from rice is distinguished from cochlioquinone A in that the methyl group is bound to c15, not c14 (Figure 5) [28]. In addition, it exhibits antifungal activity against major pathogenic fungi in rice [29]. Four phytotoxic compounds were obtained from *Bipolaris bicolor* El-1, a fungal pathogen derived from grasses, of which the compound, isocochlioquinone A, inhibits the root growth of plant seedlings, such as rice [30]. The cholesterol acyltransferase (ACAT) inhibitor, epi-cochlioquinone A, is a stereoisomer of cochlioquinone A and has been shown to absorb cholesterol in the human body [31]. Cochlioquinone was isolated from *cochliobolus miyabeanus*, a parasitic fungus of rice. *C. miyabeanus* causes apoptosis by triggering a hypersensitive response around the infection site, which limits the ability of the pathogen to spread to surrounding cells. It also triggers a defense response through the production of secondary signaling molecules including salicylic acid (SA), jasmonic acid (JA), ethylene (ETH), and nitric oxide (NO) [32]. In addition, it was predicted that the quinone component of cochlioquinone causes inhibition of redox reactions in the electron transport system [33]. Quinone-related inhibitors generally have a quinone ring or quinone-related structure at the edge of the molecule; however, the quinone moiety of cochlioquinone is located at the center of the molecule [34]. Quinones bind to leaf proteins, inhibiting protein digestion in herbivores, and negatively affect insect growth and development [35]. In the present study, when resistant and susceptible cultivars was treated with cq-9 and then inoculated with WBPH, the timing of the growth difference was different, but the growth of the plants treated with cq-9 on the 2nd and 3rd days after inoculation was superior to that of the untreated plants. Therefore, cq-9 may be related to the defense mechanism of rice against WBPH.

Many secondary metabolites, such as flavonoids, play important roles in the defense against pests. An important gene in the biosynthetic pathway of these secondary metabolites may be an alternative to enhancing rice resistance [36]. Flavonoids exhibit various pharmacological effects and play important roles in plant resistance as stress protectants, inducers, and feeding inhibitors [37]. In the present study, the relative expression of *OsF3H1*, *OsCM*, *OsWRKY45*, and *OsNPR1*, which are known defense-related genes in rice, were compared. The gene expression profiles were analyzed after treatment with cq-9 and inoculation with WBPH, and inoculation with WBPH followed by cq-9 treatment. *OsF3H* is involved in the regulation of the downstream genes, *oryza sativa dihydroflavonol 4-reductase* (*OsDFR*) and *oryza sativa flavonol synthase* (*OsFLS*), and the related genes, *SLENDER RICE1* (*OsSLR1*) and *OsWRKY13* of the flavonoid pathway. This results in the upregulation of defense-related genes and enhances rice resistance to WBPH invasion [38]. SA and JAs rapidly accumulate in plants attacked by pests and activate defense signals, which are involved in the expression of defense genes that induce defense proteins [39]. *OsCM* alleviates pathogen stress by altering the aromatic amino acid (AAA) metabolism, stress response genes, and hormone accumulation [38]. Phenylalanine (Phe), tyrosine (Tyr), and tryptophan (Trp) are major AAA metabolic molecules in plants that are responsible for the synthesis of hormones such as SA and auxin, as well as the synthesis of secondary metabolites [40]. In general, when plants are exposed to pathogens, the induction of genes associated with AAA is enhanced. Phe is used as a precursor of various phenolic compounds and is typically a precursor of flavonoids, condensed tannins, and lignin compounds [41]. The *NPR1* gene is a major regulator of SA-mediated resistance and is functionally conserved in various plant species, including rice. *NPR1* enhances resistance to various pathogens including bacteria and fungi. The function of *NPR1* has been characterized in Arabidopsis, and rice *NPR1* was recently identified. [42]. *OsNPR1* is associated with disease resistance insofar as it increases the susceptibility of herbivores and modulates resistance to WBPH [43]. WRKY belongs to a large family of transcription factors involved in plant biological stress. These proteins include a WRKY domain, a 60 amino acid region with a strongly conserved amino acid sequence, WRKYGQK, in the N-terminus [44]. Currently, 155 *OsWRKY* genes are predicted to be present in rice. WRKY transcription factors (TFs) are involved in regulating a variety of biological processes including growth, development, and biotic and abiotic stress [45]. In general, WRKY is considered to be a TF involved in various biological processes under normal or stressful conditions. After SA and pathogen infection in rice, genome-wide expression analyses were performed, which revealed that many WRKY proteins are involved in the rice defense response under such stress [46]. The relative expression levels of *OsF3H1*, *OsCM*, *OsWRKY45*, and *OsNPR1* were rapidly increased in CNDH3, CNDH42, TN1, and CNDH45 following cq-9 treatment, which demonstrates that cq-9 interacts with various defense-related genes.

## 5. Conclusions

WBPH is a major biotic stress that causes the loss of rice around the world [47]. In this study, WBPH resistance to cq-9, a bioactive substance, was evaluated and the expression profile of related genes were analyzed. After inoculating with WBPH, the effect of cq-9 on the growth of rice was determined by measuring the length of the plant. Rice inoculated with WBPH after treatment with cq-9 showed better growth compared with untreated rice. However, on the first day of inoculation, the opposite was observed. Therefore, cq-9 confers resistance to WBPH and may also affect plant growth. In addition, when the relative expression of genes involved in plant defense were compared during cq-9 treatment and WBPH inoculation, gene expression was increased on the 2nd and 3rd days following cq-9 treatment. Therefore, we confirmed that cq-9, a quinone family, interacts with plant defense genes regulating AAA and precursors of various plant secondary metabolites, and it plays a role in increasing the resistance of rice to WBPH. We suggest a novel treatment for WBPH control by utilizing cq-9 as an eco-friendly insecticide, and is also expected to contribute to the discovery of genetic resources associated with plant defense.

## Figures and Tables

**Figure 1 biology-10-01273-f001:**
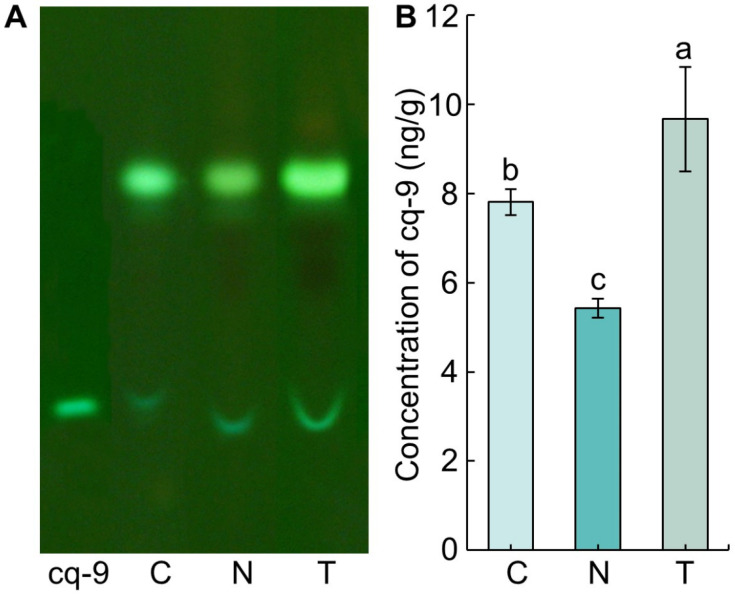
Identification of cochlioquinone-9 (cq-9) from methanol extracts of rice. (**A**) TLC results. A yellow-green band was identified at the bottom of the TLC plate. The Rf value for each band was measured, and the same Rf value indicates the same substance. (**B**) Concentration of cq-9. cq-9 was separated from the TLC silica gel. The concentration was calculated as the weight of cq-9 dried after separation [ng/fresh weight of rice (g)]. C; Cheongcheong, N; Nagdong, T; TN1, Means with the same letters were not significantly different by Duncan’s multiple range test at *p* < 0.05.

**Figure 2 biology-10-01273-f002:**
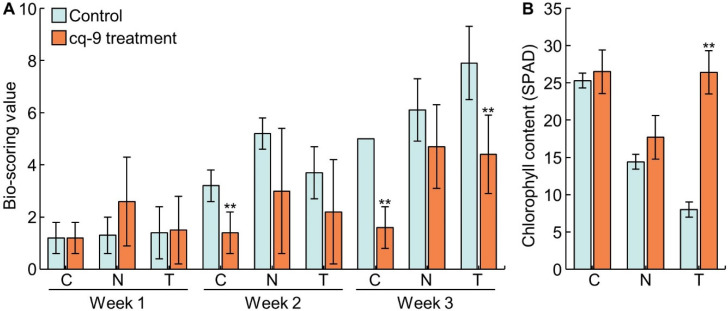
Evaluation of WBPH resistance by cultivar following cq-9 treatment. (**A**) Bio-scoring values after 1, 2, and 3 weeks after WBPH inoculation. The damage by WBPH was less when cq-9 was treated in Cheongcheong at 2 weeks, and Cheongcheong and TN1 at 3 weeks. (**B**) Chlorophyll content at 3 weeks after WBPH inoculation. C; Cheongcheong, N; Nagdong, T; TN1, ** significant difference at *p* < 0.01.

**Figure 3 biology-10-01273-f003:**
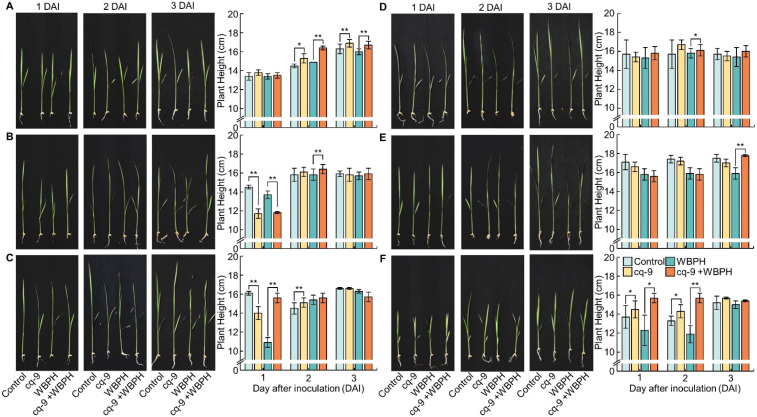
Comparison of plant length following cq-9 treatment. The control was untreated, cq-9 was treated with cq-9, WBPH was inoculated with WBPH, and cq-9 + WBPH was inoculated with WBPH following cq-9 treatment. When resistant and susceptible cultivars was treated with cq-9 and then inoculated with WBPH, the timing of the difference in growth was different, but the growth of the plants treated with cq-9 on the 2nd and 3rd days after inoculation was superior to that of the untreated plants. Therefore, cq-9 may be related to the defense mechanism of rice against WBPH and related to plant growth. (**A**) Cheongcheong. (**B**) CNDH3. (**C**) CNDH42-2. (**D**) Nagdong. (**E**) TN1. (**F**) CNDH45. DAI; day after inoculation, * significant difference at *p* < 0.05, ** significant difference at *p* < 0.01.

**Figure 4 biology-10-01273-f004:**
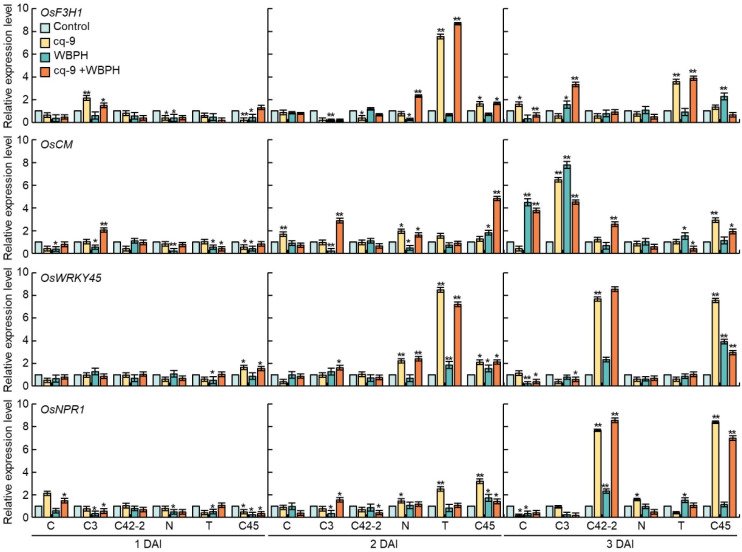
Relative expression level of plant defense-related genes, *OsF3H1*, *OsCM*, *OsWRKY45*, and *OsNPR1* following cq-9 treatment. The relative expression levels of these genes were rapidly increased in CNDH3, CNDH42, TN1, and CNDH45 at the 2nd and 3rd day after inoculation following cq-9 treatment. C; Cheongcheong, C3; CNDH3, C42-2; CNDH42-2, N; Nagdong, T; TN1, C45; CNDH45, DAI; day after inoculation, * significant difference at *p* < 0.05, ** significant difference at *p* < 0.01.

**Figure 5 biology-10-01273-f005:**
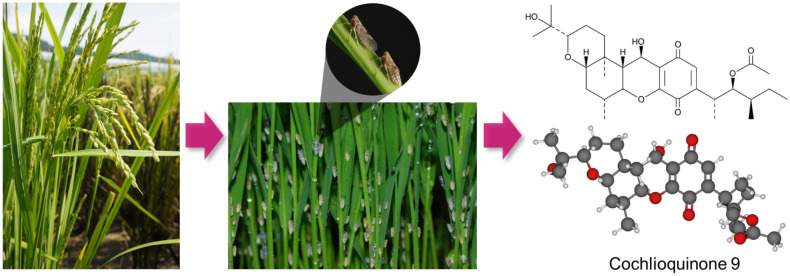
Structural comparison of cochlioquinone-9 (cq-9) isolated from rice with various cochlioquinones. Rice in the natural environment is exposed to abiotic stress caused by WBPH. WBPH-resistant rice cultivars synthesize larger amounts of cq-9 than susceptible cultivars when damaged by WBPH, and thus cq-9 is expected to contribute to plant defense. The molecular formula of cq-9 is (3R)-9-[(1S,2R,3S)-2-acetyloxy-1,3-dimethylpentyl]-1,2,3,4aβ,5,6,6a,12,12aβ,12b-decahydro-12β-hydroxy-3α-(1-hydroxy-1-methylethyl)-6aα,12bα-dimethylpyrano [3,2-a]xanthene-8,11-dione (C_30_H_44_O_8_), and the molecular weight is 532.7 g/mol. cq-9 is distinguished from cochlioquinone A in that c26 is bound not to c14 but to c15.

**Table 1 biology-10-01273-t001:** Identification and separation of cq-9 using TLC.

	Control	Cultivars			*p* Value
		Cheongcheong	Nagdong	TN1	
Rf value	0.43 ± 0.01 a ^z^	0.42 ± 0.01 a	0.42 ± 0.01 a	0.42 ± 0.01 a	NS
Concentration of cq-9 (ng/g)	-	7.81 ± 0.29 b	5.43 ± 0.21 c	9.67 ± 1.17 a	<0.001 **

^z^ The data are presented as the mean ± standard deviation. Means with the same letters are not significantly different by Duncan’s multiple range test at *p* < 0.05. ** significant at the 0.01 level.

**Table 2 biology-10-01273-t002:** Measurement of WBPH resistance according to cq-9 treatment.

	Week ^z^	Cultivars	Control	cq-9 Treatment	*p* Value
Bio-scoring value	1	Cheongcheong	1.2 ± 0.6 ^y^	1.2 ± 0.6	0.172
		Nagdong	1.3 ± 0.7	2.6 ± 1.7	0.124
		TN1	1.4 ± 1.0	1.5 ± 1.3	0.304
	2	Cheongcheong	3.2 ± 0.6	1.4 ± 0.8	0.004 **
		Nagdong	5.2 ± 0.6	3.0 ± 2.4	0.072
		TN1	3.7 ± 1.0	2.2 ± 2.0	0.051
	3	Cheongcheong	5.0 ± 0.0	1.6 ± 0.8	0.008 **
		Nagdong	6.1 ± 1.2	4.7 ± 1.6	0.099
		TN1	7.9 ± 1.4	4.4 ± 1.5	0.008 **
Chlorophyll content (SPAD)	3	Cheongcheong	5.0 ± 0.0	1.6 ± 0.8	0.057
		Nagdong	6.1 ± 1.2	4.7 ± 1.6	0.379
		TN1	7.9 ± 1.4	4.4 ± 1.5	0.002 **

^z^ Number of weeks after WBPH inoculation. ^y^ The data are presented as the mean ± standard deviation. ** significant difference at *p* < 0.01.

**Table 3 biology-10-01273-t003:** Plant length according to WBPH inoculation and cq-9 treatment.

Cultivars	DAI ^z^	Control	cq-9	*p* Value	WBPH	cq-9 + WBPH	*p* Value
Cheongcheong	1	13.4 ± 0.4 ^y^	13.8 ± 0.1	0.060	13.4 ± 0.3	13.5 ± 0.3	0.585
	2	14.5 ± 0.2	15.3 ± 0.5	0.025 *	14.9 ± 0.07	16.4 ± 0.2	0.007 **
	3	16.3 ± 0.5	16.9 ± 0.4	0.007 **	16.0 ± 0.3	16.7 ± 0.4	0.0004 **
CNDH 3	1	14.5 ± 0.2	11.7 ± 0.5	0.003 **	13.7 ± 0.4	11.8 ± 0.1	0.0006 **
	2	15.8 ± 0.7	16.1 ± 0.5	0.237	15.8 ± 0.6	16.4 ± 0.5	0.006
	3	15.9 ± 0.3	15.8 ± 0.7	0.618	15.7 ± 0.4	15.9 ± 0.6	0.836
CNDH 42-2	1	16.1 ± 0.2	14.0 ± 0.7	0.008 **	10.9 ± 0.5	15.6 ± 0.5	0.005 **
	2	14.5 ± 0.6	15.1 ± 0.5	0.003 **	15.4 ± 0.5	15.6 ± 0.5	0.096
	3	16.6 ± 0.1	16.6 ± 0.1	0.688	16.3 ± 0.2	15.7 ± 0.5	0.062
Nagdong	1	15.7 ± 1.5	15.4 ± 0.5	0.549	15.3 ± 1.1	15.8 ± 0.7	0.833
	2	15.7 ± 1.5	16.7 ± 0.5	0.144	15.8 ± 0.5	16.1 ± 0.6	0.031 *
	3	15.7 ± 1.6	15.5 ± 0.5	0.760	15.4 ± 1.0	16.0 ± 0.6	0.596
TN1	1	17.1 ± 0.8	16.6 ± 0.5	0.160	15.8 ± 0.6	15.6 ± 0.6	0.228
	2	17.4 ± 0.4	17.2 ± 0.4	0.372	15.9 ± 0.6	15.8 ± 0.6	0.146
	3	17.5 ± 0.4	1.7 ± 0.4	0.077	15.9 ± 0.6	17.8 ± 0.1	0.002 **
CNDH 45	1	13.7 ± 1.2	14.5 ± 0.9	0.037 *	12.3 ± 1.6	15.7 ± 0.5	0.027 *
	2	13.3 ± 0.5	14.3 ± 0.7	0.041 *	11.9 ± 0.9	15.7 ± 0.5	0.008 **
	3	15.2 ± 0.7	15.7 ± 0.1	0.180	15.0 ± 0.4	15.4 ± 0.1	0.584

^z^ DAI means days after inoculation. ^y^ The data are presented as the mean ± standard deviation. * significant difference at *p* < 0.05, ** significant difference at *p* < 0.01.

## Data Availability

Not applicable.

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
