# Peer review of "Biological Efficacy of Cochlioquinone-9, a Natural Plant Defense Compound for White-Backed Planthopper Control in Rice"

_biology, 2021, doi:10.3390/biology10121273_

Round 1
Reviewer 1 Report
In general, the structure of the ms should be improved. The abstract needs to be improved as well.
I consider that conclusions should be referenced.
See comments attached in the ms.

Reviewer 2 Report
Submitted MS “Biological Efficacy of Cochlioquinone-9, a Natural Plant Defense Compound for White-backed Planthopper Control in 3 Rice” by Jang et al., reports the improved plant ability to resist WBPH after the treatment with cochlioquinone-9 (cq-9), a compound from quinone family and attempted to demonstrate mechanism behind this phenomenon. The specific comments:
- The introduction part needs substantial improvements. Authors should incorporate the research developments and available measures to control planthoppers.
- L 90-185: Authors have described several numerical values with standard deviation throughout these sections. I would suggest to provide data in table form and describing the conclusive results from the experiments.
- The table legends can be described with more clarity. The statistical parameters are mostly unclear.
- Fig 5: Figure legend tells about the cochlioquinone-9 but why do we see the grasshoppers? No mention of rice and grasshopper image.
- Authors need to provide the number of technical and biological replicates for each experiment and mention the number of times, each experiment was repeated with similar results. They need to provide the statistical method employed to calculate p-values.
- The authors have used the term “resistant” against WBPH. Please make sure to use the correct term. Apparently, data display “tolerant” against WBPH.
- Fig. 3B, C, G: On the first day of treatment/inoculation, the plant shows significant differences in height but not at day 3. It suggests the plants used for the study were not uniform. This could be the reason for variable tolerant response against WBPH.
Round 2
Reviewer 1 Report
I consider acceptable the changes performed.
Reviewer 2 Report
The authors have addressed most of my concerns appropriately.